# Chemical Instability-Induced Wettability Patterns on Superhydrophobic Surfaces

**DOI:** 10.3390/mi15030329

**Published:** 2024-02-27

**Authors:** Tianchen Chen, Faze Chen

**Affiliations:** 1School of Mechanical Engineering, Tianjin University, Tianjin 300350, China; chentianchen@tute.edu.cn; 2Laboratory of Packaging Engineering and Visual Interaction Design, Tianjin University of Technology and Education, Tianjin 300222, China

**Keywords:** superhydrophobic surface, chemical instability, wettability pattern, droplet manipulation

## Abstract

Chemical instability of liquid-repellent surfaces is one of the nontrivial hurdles that hinders their real-world applications. Although much effort has been made to prepare chemically durable liquid-repellent surfaces, little attention has been paid to exploit the instability for versatile use. Herein, we propose to create hydrophilic patterns on a superhydrophobic surface by taking advantage of its chemical instability induced by acid solution treatment. A superhydrophobic Cu(OH)_2_ nanoneedle-covered Cu plate that shows poor stability towards HCl solution (1.0 M) is taken as an example. The results show that 2.5 min of HCl solution exposure leads to the etching of Cu(OH)_2_ nanoneedles and the partial removal of the self-assembled fluoroalkyl silane molecular layer, resulting in the wettability transition from superhydrophobocity to hydrophilicity, and the water contact angle decreases from ~160° to ~30°. Hydrophilic dimples with different diameters are then created on the superhydrophobic surfaces by depositing HCl droplets with different volumes. Afterwards, the hydrophilic dimple-patterned superhydrophobic surfaces are used for water droplet manipulations, including controlled transfer, merging, and nanoliter droplet deposition. The results thereby verify the feasibility of creating wettability patterns on superhydrophobic surfaces by using their chemical instability towards corrosive solutions, which broadens the fabrication methods and applications of functional liquid-repellent surfaces.

## 1. Introduction

Bio-mimic superhydrophobic surfaces have drawn rapidly increasing research interest because of their great significance for versatile applications such as self-cleaning [1,2], chemical detection and sensors [3,4,5,6], liquid droplet/gas bubble manipulation [7,8,9,10,11], icephobicity [12,13,14,15,16], immiscible liquid separation [17,18,19], enhanced heat transfer [20,21,22], and anti-corrosion [23,24,25]. However, most superhydrophobic surfaces suffer from chemical instability in harsh conditions like highly concentrated acidic/alkaline/salty solution immersion, organic solvent invasion, thermal treatment, UV irradiation, and active species exposure [26,27,28]. The chemical instability usually results in the loss of superhydrophobicity to varying degrees due to interfacial chemical process-induced surface chemistry and/or morphology change, which could obviously shorten the lifespan of superhydrophobic surfaces and thus be commonly considered one of the major limitations for their real-world applications. As a result, great efforts have been made to fabricate chemically durable superhydrophobic surfaces, and much progress has been achieved in recent years [27,28].

Although chemically stable superhydrophobic surfaces are crucial in some cases, such as chemical shielding and anti-corrosion [29,30,31], external stimuluses triggered chemical instabilities are sometimes more desired, especially in the construction of functional surfaces with (super)hydrophilic–superhydrophobic patterns or reversible wettability [28,32,33,34]. Surfaces with extreme wettability patterns are widely used in various applications such as enhanced water harvesting, spontaneous liquid transport, etc. [28,35]. For example, Bai et al. [36] took the advantage of the chemical instability of superhydrophobic TiO_2_ coatings upon UV exposure to prepare star-shaped superhydrophilic patterns on superhydrophobic surfaces, which were used to realize efficient water harvesting by integrating the water collection strategies of both desert beetles and spider silk. By using a similar patterning method, Ghosh et al. [37] prepared wedge-shaped superhydrophilic patterns on superhydrophobic TiO_2_ surfaces and realized high-rate, pumpless liquid transport on the open substrates. Xu et al. [38] reported selective UV irradiation-induced superhydrophilic patterns on an octadecytrichlorosilane-modified superhydrophobic silica coating and demonstrated its potential application towards microgravity biosensing. Plasma contains active species, such as high-energy electrons, metastable particles, etc., so it can always initiate surface chemistry and/or texture changes and thus, cause instability (i.e., hydrophilization) of superhydrophobic surfaces [32]. For example, Huang et al. [39] used the chemical instability of a superhydrophobic surface towards plasma treatment to fabricate superhydrophilic patterns on the superhydrophobic surface, which was further used for underwater spontaneous pumpless transportation of organic liquids. Liu et al. [40] fabricated hydrophilic patterns on a superhydrophobic surface by micro-plasma jet treatment and reported its versatile ability to control water adhesion. Additionally, Liu et al. [41] showed that a superhydrophobic Cu mesh could be completely converted to be superhydrophilic after being immersed in tetrahydrofuran for 5 min, which facilitated subsequent reversible oil/water separation. Therefore, the chemical instability of superhydrophobic surfaces towards external stimuluses is not always a drawback for their applications; making ingenious use of the instability can provide additional possibilities for constructing special functional surfaces [28].

Immersion in an acidic/alkaline/salty solution, triggering chemical corrosion, can sometimes change the surface morphology and/or surface chemistry of superhydrophobic surfaces, thereby changing their surface wettability (i.e., chemical instability) [42,43,44,45]. Although some works involving pH-responsive switchable super-wettability have been reported [43,44,45], they were limited to smart surface fabrication or reversible oil/water separation, indicating that the full exploitation of their chemical instability for beneficial usage was largely insufficient. In this paper, we took a superhydrophobic Cu(OH)_2_ nanorod-covered Cu surface, which was unstable towards HCl solution exposure, as an example to demonstrate the feasibility of creating wettability patterns by using this chemical instability. Chemical etching by an HCl solution (1.0 M, 2.5 min) turned the superhydrophobic surface into a hydrophilic one, and the related mechanism of the wettability transition was studied. Hydrophilic dimple-patterned superhydrophobic surfaces were then prepared and employed to realize water droplet manipulation, such as transfer, merging, and deposition.

## 2. Materials and Methods

Copper plates (3 × 4 × 0.1 cm^3^) were bought from Huaru copper Co., Ltd. (Guangzhou, China). Analytical-grade ethanol, HCl, NaOH, and (NH_4_)_2_S_2_O_8_ were supplied by Tianjin Kemiou Chemical Reagent Co., Ltd. (Tianjin, China). 1H,1H,2H,2H-Perfluorodecyltriethoxysilane (fluoroalkyl silane, FAS) with 97% purity was purchased from Alfa Aesar (Haverhill, MA, USA).

A superhydrophobic surface on a copper substrate was fabricated according to a previously reported method [46]. Briefly, a copper plate was firstly polished mechanically using 1000# and 2000# abrasive paper and then ultrasonically cleaned in sequence in HCl (0.1 M), alcohol, and deionized water. Then, the cleaned copper plate was placed into an aqueous solution containing NaOH (2.5 M) and (NH_4_)_2_S_2_O_8_ (0.1 M) for 5 min to construct micro/nano structures. Then, the substrate was taken out and washed with abundant ultrapure water. After drying with blowing air, the sample was immersed into 1 wt% ethanol solution of FAS for 1 h to lower the surface energy. Then, the plate was washed with ethanol and dried at 90 °C. Finally, the copper surface was imparted with superhydrophobicity.

The instability of the obtained superhydrophobic Cu surface was triggered by an HCl (1.0 M) solution, and the chemical instability was characterized by the water contact angles (WCAs) of the surface after being exposed to the HCl solution for different times (0∓5 min). Hydrophilic dimple patterns were fabricated by depositing HCl (1.0 M) droplets with different volumes (0.2∓5.0 μL) on the as-prepared superhydrophobic Cu surfaces for 2.5 min. After reaching the specific reaction time, the residual liquid was removed by absorbent paper, and the surfaces were washed with deionized water and then blow-dried at room temperature.

The WCAs of the sample surfaces were measured by an optical contact angle meter (AST-VCM Optima, Billerica, MA, USA) at room temperature. The 3D morphology and corresponding cross-sectional profile were recorded by a confocal laser scanning microscope (CLSM, Carl Zeiss LSM 700, Jena, Germany). The surface morphologies and corresponding chemical compositions of the samples were observed by scanning electron microscope (SEM, SUPRA 55 SAPPHIRE, Oberkochen, Germany) at an accelerating voltage of 15 kV. The surface chemical groups were analyzed by Fourier transform infrared spectrophotometer (FTIR, JASCO, Tokyo, Japan). The surface chemistries were also characterized by X-ray photoelectron spectroscopy (XPS, Thermo ESCALAB 250Xi, Waltham, MA, USA) with a monochromatic Al Kα (1486.6 eV) X-ray beam, and the C 1s peak at 284.8 eV was used as reference. The spot size was 400 μm, and the pass energy and energy step size of the full spectrum scan were, respectively, 100 eV and 1.0 eV, while those of the C 1s high-resolution spectrum were 50.0 eV and 0.1 eV, respectively. An X-ray diffractometer (XRD, Bruker AXS D8 Discover, Bremen, Germany) with an X-ray source of Cu Kα (λ = 1.5418 Å) was used to observe the crystal-phase structure of the samples; the scanning rate was 2°/min.

## 3. Results and Discussion

The as-prepared superhydrophobic Cu surface possessed excellent water repellence with a water contact angle (WCA) of 160°, and the solid–liquid interfacial adhesion was negligible when it was pressed to make contact with a water droplet (5.0 μL), as shown in Figure 1a. By contrast, as illustrated in Figure 1b, if the superhydrophobic substrate was moved up and contacted an HCl droplet (1.0 M, 5.0 μL), obvious interfacial adhesion could be observed when the substrate was moved down, and the HCl droplet was even dragged down from the liquid-feeding outlet and then tightly adhered on the surface, demonstrating locally damaged water repellence of the surface upon HCl droplet exposure. The HCl solution-induced chemical instability of the superhydrophobic surface was further studied by measuring the WCA of samples that were immersed in HCl for different durations. Figure 1c shows the influence of immersion time on the WCA of the superhydrophobic surface. It can be seen that the WCA decreased with the immersion time in HCl, and the surface lost its superhydrophobicity after several seconds of chemical etching. Finally, the WCA stabilized at ~30° after immersion for 150 s, the surface turned its color to bronze, and the deposited water droplet spread rather than beaded up on the etched surface, as depicted in Figure 1d, indicating complete loss of the water repellence by exposing the surface to the corrosive HCl solution.

Taking advantage of the HCl-induced chemical instability of the superhydrophobic surfaces, hydrophilic patterns can be constructed on the surfaces. Here, we propose to create hydrophilic dimple patterns on superhydrophobic Cu surfaces by depositing HCl droplets on the surface to trigger localized chemical etching, as illustrated in Figure 2a. Figure 2b–f show the surface morphology characterizations of the dimple pattern prepared by using a HCl droplet with volume of 0.2 μL. As depicted in Figure 2b,c, the CSML image and the corresponding cross-sectional profile clearly showed that a circular dimple with diameter of ~390 μm and depth of ~3 μm was obtained on the superhydrophobic surface. It is widely known that when a water droplet is deposited on a superhydrophobic surface, a circular liquid–solid–vapor composite contacting the interface can be built. Here, the placed HCl droplet-triggered chemical corrosion occurred at the HCl-Cu(OH)_2_ contact area and finally created a dimple pattern with a round shape. Figure 2d–f show SEM images of the hydrophilic dimple-patterned superhydrophobic surface with different magnifications. It can be clearly seen that the as-prepared superhydrophobic Cu surface (i.e., the unetched area in Figure 2d) was covered with numerous nanorods and some flower-like microstructures, as can be observed in Figure 2e. By contrast, after being etched by an HCl droplet, the nanorods and flower-like microstructures were obviously destroyed and replaced by closely packed irregular particles with sizes in a range from hundreds of nanometers to several micrometers (Figure 2f). Zooming into these irregular particles revealed that they were decorated with nano-scale granules.

Figure 3 shows the surface chemistries of the original and HCl-etched areas on the superhydrophobic Cu surface. Figure 3a shows the FTIR spectra of the two areas, and the bands around 1139, 1211, and 1241 cm^−1^ were attributed to the –CF stretching vibrations of the –CF_2_ and –CF_3_ groups [47]. The weak bands at 779 cm^−1^ were assigned to the Si–O–CH_2_CH_3_ groups of the FAS molecules [48], while the emergence of peaks around 894 cm^−1^ were related to the stretching vibrations of the Si–O–Si bonds [47]. The Si–O–CH_2_CH_3_ groups in the FAS molecule were transformed into Si–OH groups due to the hydrolysis reaction in ethanol solution [49,50]. When the rough Cu substrate was immersed in the ethanol solution of FAS, a second hydrolysis reaction between the Si–OH groups and the surface –OH groups occurred to form Si–O–Cu bonds, which enabled successful self-assembly of the FAS molecules on the Cu surface. Some adjacent self-assembled FAS molecules underwent further dehydration and condensation reactions to form Si–O–Si bonds, leading to the formation of a denser fluorosilane network to impart the rough Cu surface with superhydrophobicity [48,49]. It can be observed that after HCl etching, the intensities of these peaks decreased, especially those originating from the FAS molecules, indicating that the self-assembled FAS coating was partially removed.

Figure 3b depicts the XPS spectra of the superhydrophobic and HCl-etched areas, and elements of Cu, C, O, F, and Si could be detected. The relative atomic percent of element F was measured to be 20.8% at the original superhydrophobic area, which originates from FAS molecules and contributes to the formation of superhydrophobicity. After HCl etching, the surface F decreased to 4.17%, Cu reduced from 14.74% to 7.61%, and O decreased from 26.74% to 15.78%. Notably, Cl with a content of 4.97% was detected in the HCl-etched area. According to the high-resolution C 1s spectra shown in Figure 3c, the peaks assigned to the −CF_2_ and −CF_3_ groups of the FAS molecules were markedly detected on the superhydrophobic area [31], while the corresponding peaks for the HCl-etched area could hardly be observed; that is, the peak intensities were greatly weakened after HCl-induced corrosion. Figure 3d shows the XRD patterns of the original superhydrophobic area and HCl-etched area. It can be seen that, besides the diffraction peaks at 2θ = 43.3, 50.4, 74.1, and 89.9°, respectively attributed to face-centered cubic Cu planes of (111), (200), (220), and (311) (JCPDS card No. 04-0836) that originated from the Cu substrate, new orthorhombic-phase Cu(OH)_2_ planes of (021) at 23.7°, (002) at 34.0°, (111) at 35.8°, (041, 022) at 38.0°, (130) and 39.7°, and (150, 132) at 53.2° (JCPDS card No. 80-0656) were detected, confirming the well-known alkali-assisted oxidation of Cu to generate Cu(OH)_2_ [46,51]:Cu + 4NaOH + (NH_4_)_2_S_4_O_8_ → Cu(OH)_2_ + 2Na_2_SO_4_ + 2NH_3_↑ + 2H_2_O(1)

In comparison, these diffraction peaks of Cu(OH)_2_ disappeared after HCl etching, while three new characteristic peaks corresponding to CuCl planes of (111) at 28.6°, (220) at 47.5°, and (311) at 56.3° (JCPDS card No. 06-0344) were observed. It is commonly known that neutralization happens when Cu(OH)_2_ contacts HCl, as described in the following equation:Cu(OH)_2_ + 2HCl → CuCl_2_ + 2H_2_O(2)

Then, the generated CuCl_2_ could partly etch the exposed Cu substrate with the assistance of HCl, which was finally reduced to CuCl and precipitated on the Cu substrate [52,53]:CuCl_2_ + Cu → 2CuCl↓(3)

According to the above results, we could conclude that NaOH-assisted surface oxidation generated Cu(OH)_2_ micro/nano rough structures on the Cu substrate, and the subsequent immersion in FAS successfully triggered the formation of an F-containing monolayer with low surface tension; the obtained surface thereafter possessed superhydrophobicity [46,54]. However, when HCl solution invaded, the Cu(OH)_2_ nanorods were chemically etched and partially washed away, as well as the self-assembled FAS layer, which resulted in the content decrease in surface Cu, O, and F. Therefore, the HCl-etched area lost its superhydrophobicity and showed hydrophobicity, which enabled us to construct hydrophobic patterns on the superhydrophobic Cu surface by using its chemical instability towards HCl.

We then examined the influence of the volume of the deposited HCl droplet (*V*_HCl_, 0.2–5.0 μL) on the diameter of the obtained dimple (*D*); the results are depicted and fitted in Figure 4. It can be seen that the diameter of the obtained dimple grew almost linearly with the increase in *V*_HCl_. For example, when the *V*_HCl_ was as small as 0.2 μL, a dimple with a diameter of 0.38 ± 0.02 mm could be obtained. Smaller HCl droplets could hardly generate observable dimples due to their rapid evaporation. When the *V*_HCl_ was increased to 1.0 μL, the dimple diameter was about 0.47 ± 0.01 mm, and it increased to 0.91 ± 0.03 mm when the *V*_HCl_ was 5.0 μL. According to the linear fitting of the experimental data (the correlation coefficient *R*^2^ = 0.99), the dimple diameter, *D*, could be estimated as follows:*D* = 0.375 + 0.111*V*_HCl_(4)

Programmable liquid droplet manipulations, including controllable transfer and deposition, have been widely explored on wettability-patterned surfaces [55,56,57,58]. Here, we tested water droplet manipulations on the HCl etching-patterned hydrophilic/superhydrophobic surface. Figure 5 shows water droplet transfer by hydrophilic dimple-patterned superhydrophobic substrates. Figure 5a illustrates the schematic diagram of the pre-designed patterned surfaces, consisting of one hydrophilic dimple on each superhydrophobic substrate. As shown in the image in Figure 5b, a water droplet (5 μL) was initially pre-deposited at position A of the superhydrophobic surface; positions B and C were patterned with hydrophilic dimples with diameters of 0.62 mm and 0.81 mm, respectively. Figure 5c,d depict that an upward movement of the lower substrate enabled the contact between the droplet and dimple B, and then the droplet adhered to the upper substrate via dimple B when the lower plate was moved down. As can be seen in Figure 5e–h, when the hanging droplet came in contact with dimple C on the lower plate, it was grabbed by the dimple and then re-deposited onto the lower substrate, i.e., the droplet was transferred from position A to C. It could be observed that some liquid was left on the hydrophilic dimple B after the transfer, and its volume was estimated to be ~0.03 μL. Although the liquid deposition by the hydrophilic dimple was inevitable, the as-prepared patterned surface herein could be used for water droplet transfer after taking the dimple-diameter-dependent and quantifiable volume loss into consideration.

Figure 6 shows water droplets merging by using the as-prepared hydrophilic dimple-patterned superhydrophobic surfaces. As can be seen from Figure 6a, water droplets #1 (7.0 μL) and #2 (4.0 μL) were, respectively, deposited at position A (superhydrophobic area) and position B (a hydrophilic dimple with diameter of 0.81 mm); positions C and D on the upper plate were a hydrophilic dimple (diameter 0.62 mm) and a superhydrophobic area, respectively. Figure 6b–d show that after the two droplets came in contact with the upper plate at positions C and D, the subsequent downward movement of the lower plate enabled the capture of droplet #1 by the upper plate due to the high adhesion of dimple C, while droplet #2 remained at dimple B because the adhesion of position D, the original superhydrophobic surface, was extremely low. Then, horizontal and subsequently perpendicular movement of the lower substrate enabled the merging of droplets #1 and #2, resulting in the formation of a new droplet #(1 + 2) at dimple B, as depicted in Figure 6d–f. It has to be noted that a slight volume loss (~0.03 μL) of droplet #1 was observed at position C during the merging, which should be taken into consideration in terms of quantitative droplet merging. Consequently, we could envision that the proposed chemical instability-induced patterned superhydrophobic surfaces could be potentially used for droplet-based micro-reactors [59].

The deposition of nanoliter-sized droplets is important in many chemical and biomedical droplet-based microfluidic applications, such as fluorescence detection [55] and high-throughput cell screening [60]. We show here the nanoliter water deposition by the as-prepared hydrophilic dimple-patterned superhydrophobic surface, which is schematically illustrated in Figure 7a. As can be seen in Figure 7b, a water droplet (1.5 μL) attached to a needle was initially positioned on the superhydrophobic area of the patterned surface. When the substrate was horizontally moved with a velocity of ~0.6 mm/s, the droplet remained quasi-spherical in shape until it adhered to the dimple. Afterwards, the droplet gradually deformed and broke, leaving a tiny droplet deposited on the dimple. According to the schematic diagram in Figure 7c, the deposited water could be roughly considered as a spherical crown intercepted from a sphere with radius of *R* to estimate its volume. The intercepted area’s diameter (*d*) and height (*h*) of the spherical crown could be experimentally measured from the recorded images. According to the Pythagorean theorem, we know that:(5)$$x^{2}+y^{2}=R^{2}$$
(6)$${\left(R-h\right)}^{2}+{\left(d/2\right)}^{2}=R^{2}$$

The volume of the deposited water, *V*_d_, can be described as:(7)$$V_{d}=\int_{R-h}^{R}\pi x^{2}dy$$

According to Equations (5)–(7), the volume of the deposited water can be expressed as:(8)$$V_{d}=\frac{\pi h(3d^{2}+4h^{2})}{24}$$

According to Equation (8), the deposited water droplet in Figure 7b was calculated to be ~20 nL. Then, we prepared hydrophilic dimples with different diameters by using HCl droplets with varying volumes (Figure 4) and studied the influence of the dimple diameter on the volume of the deposited droplet. As shown in Figure 7d, the volume of the deposited droplet increased with the growth in dimple diameter within the examined regime, and a volume as low as 10.5 ± 1.1 nL could be deposited on the dimple with a diameter of 0.37 ± 0.02 mm. Besides droplet-based microfluidic applications, quantifiable droplet deposition can also be used to estimate the liquid loss during droplet transfer and merging, as depicted in Figure 5 and Figure 6.

## 4. Conclusions

In summary, we reported a simple method to create hydrophilic patterns on a superhydrophobic Cu surface by using HCl solution-triggered chemical instability. A 2.5 min HCl exposure led to the etching of surface micro/nano structures and partial removal of the FAS molecular layers, enabling the treated superhydrophobic area to be hydrophilic. Based on this HCl-induced chemical instability, we prepared hydrophilic dimples with diameters of 0.38–0.91 mm by depositing HCl droplets with volumes of 0.2–5.0 μL on the superhydrophobic surfaces. Typical applications, such as controlled droplet transfer, merging, and nanoliter droplet deposition, have been demonstrated on the hydrophilic dimple-patterned superhydrophobic surface. These results demonstrated the feasibility of creating wettability patterns on superhydrophobic surfaces by using their chemical instability towards corrosive solutions, which could enrich the fabrication methods of liquid-repellent surfaces with patterned wettability and promote their applications in droplet manipulation. 

## Figures and Tables

**Figure 1 micromachines-15-00329-f001:**
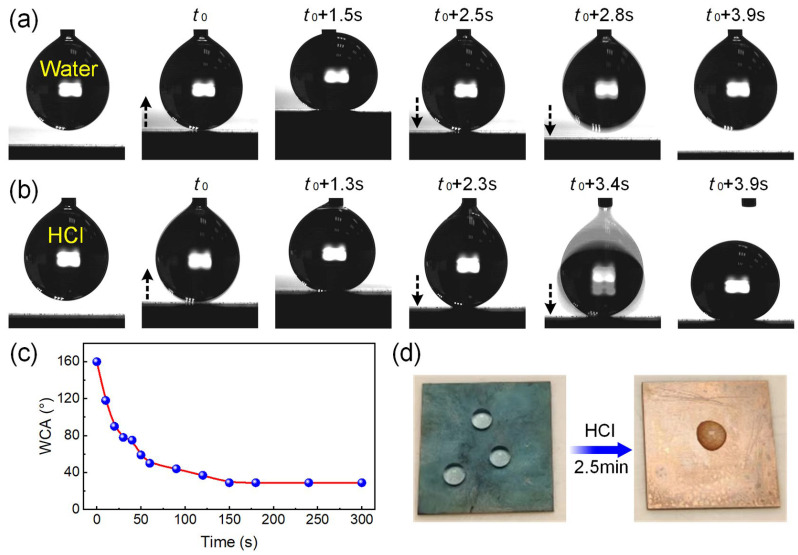
Chemical instability of the superhydrophobic Cu substrate towards HCl solution: image sequences showing the dynamic processes of (**a**) a water droplet (5.0 μL) and (**b**) an HCl droplet (1.0 M, 5.0 μL) contacting and detaches from the superhydrophobic surfaces, the arrows indicate the moving directions of the superhydrophobic surfaces; (**c**) the influence of immersion time in HCl solution on the WCA of the superhydrophobic surface; (**d**) digital image of water droplets on the original and HCl-etched (for 2.5 min) superhydrophobic surface.

**Figure 2 micromachines-15-00329-f002:**
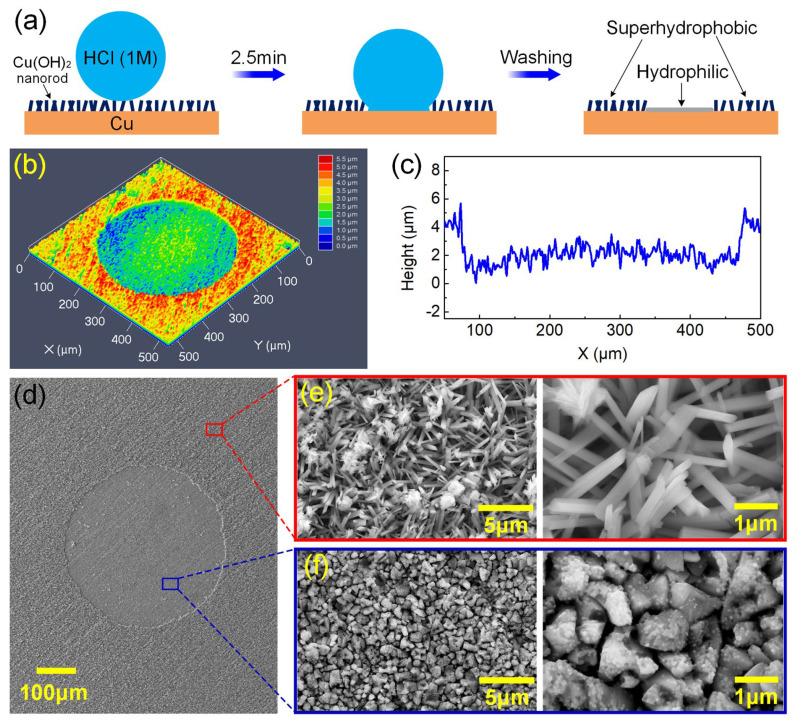
Surface morphologies of the HCl droplet (0.2 μL)-etched superhydrophobic surface: (**a**) the CLSM image and (**b**) the corresponding cross-sectional profile; (**c**–**f**) SEM images with different magnifications.

**Figure 3 micromachines-15-00329-f003:**
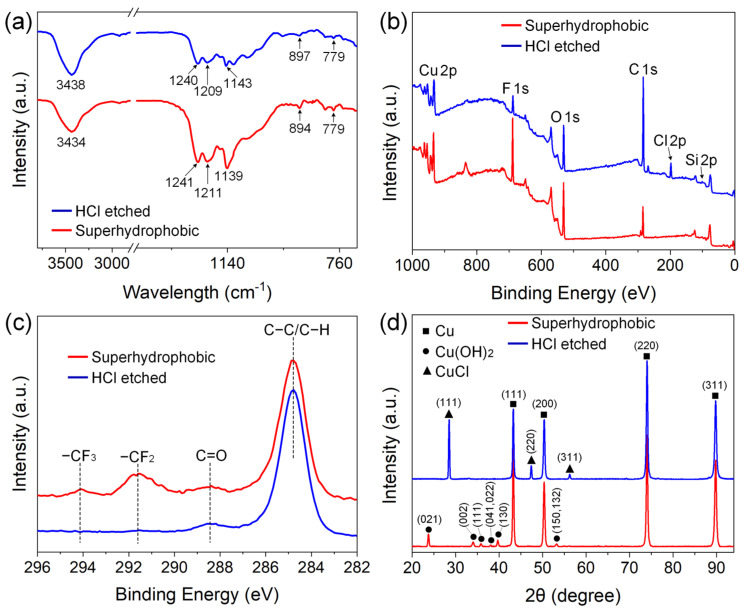
Surface chemistries of the original and HCl-etched areas on superhydrophobic Cu surface: (**a**) FTIR spectra; (**b**) XPS spectra; (**c**) high-resolution C 1s spectra; (**d**) XRD patterns.

**Figure 4 micromachines-15-00329-f004:**
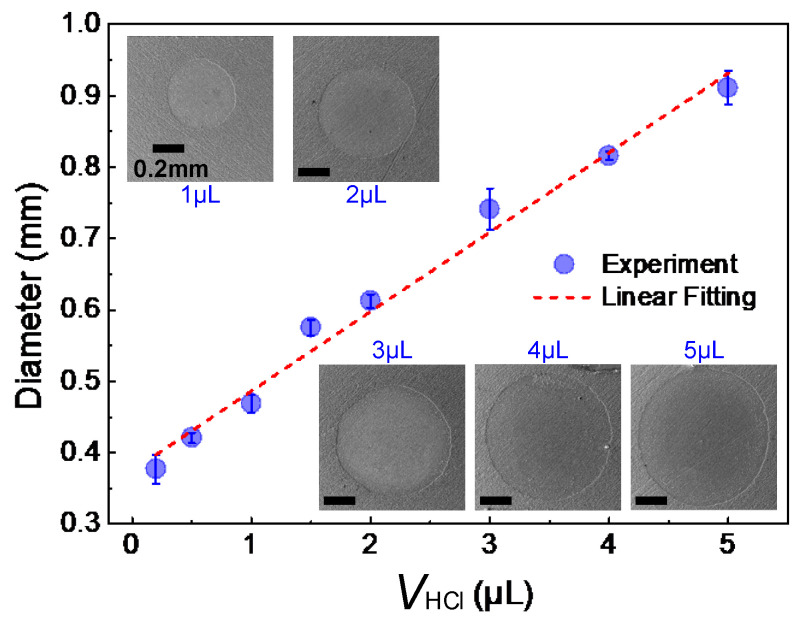
Relationship between the volume of deposited HCl droplet (*V*_HCl_) and the diameter of the obtained dimple. Spots are the experimental values, while the dotted line is their linear fitting (*R*^2^ = 0.99), and the inserted SEM images show the prepared dimples with different *V*_HCl_.

**Figure 5 micromachines-15-00329-f005:**
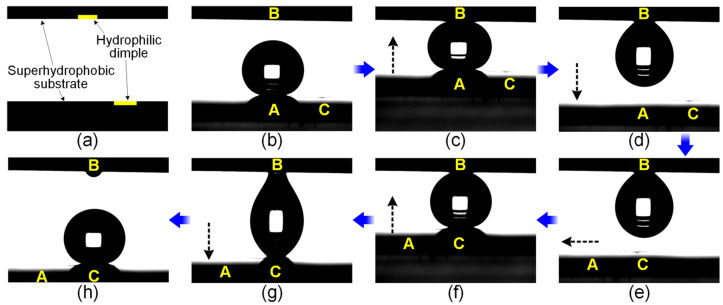
Water droplet transfer by hydrophilic dimple-patterned superhydrophobic substrates: (**a**) the schematic diagram of the employed substrates; (**b**–**h**) the image sequences showing the water droplet (5 μL) transfer from position A to C via dimple B; the black dotted arrows indicate the direction of movement of the lower substrate.

**Figure 6 micromachines-15-00329-f006:**
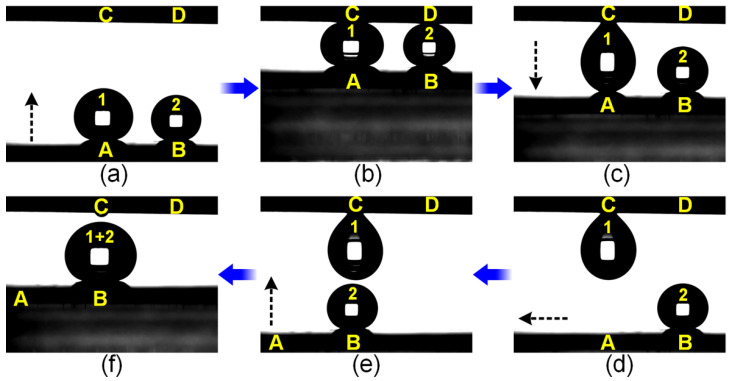
(**a**–**f**) Merging of water droplets 1 and 2 by hydrophilic dimple-patterned superhydrophobic substrates; the black dotted arrows indicate the moving direction of the lower substrate, A and B respectively indicate the superhydrophobic position and the hydrophilic dimple with diameter of 0.81 mm on the lower substrate, C and D represent the hydrophilic dimple with diameter of 0.62 mm and the superhydrophobic area on the upper substrate, respectively.

**Figure 7 micromachines-15-00329-f007:**
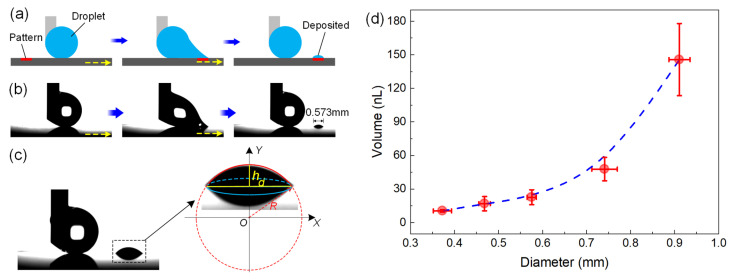
Nanoliter water deposition on hydrophilic dimple-patterned superhydrophobic substrates: (**a**) the schematic diagram and (**b**) image sequences showing a nanoliter water deposition process; the yellow arrows indicate the direction of movement of the substrate; (**c**) the schematic diagram of volume estimation for the deposited water; (**d**) the relationship between the diameter of hydrophilic dimple and the volume of deposited water droplet.

## Data Availability

The data presented in this study are available on request from the corresponding author. The data are not publicly available due to a confidentiality request.

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
