# Peer review of "Chemical Instability-Induced Wettability Patterns on Superhydrophobic Surfaces"

_micromachines, 2024, doi:10.3390/mi15030329_

Round 1

Reviewer 1 Report

Comments and Suggestions for Authors
  1. 1.What is the main challenge discussed in the research work regarding the real-world applications of liquid-repellent surfaces, and how does chemical instability contribute to this challenge?

  2. 2. How is the chemical instability of a superhydrophobic Cu(OH)2 nanoneedle-covered Cu plate exploited in the proposed method, and what specific treatment is applied to induce the instability?

  3. 3. What are the observable changes in the superhydrophobic surface after exposure to the 1.0 M HCl solution, and how do these changes lead to a transition in wettability?

  4. 4. Hydrophilic dimples with different diameters are created on the superhydrophobic surfaces, including the role of HCl droplets with varying volumes. Methodology can be included breifly

  5. 5.What is the impact of the chemical instability-induced wettability transition on the water contact angle, and how does it change from superhydrophobicity to hydrophilicity?

  6. 6.How are the hydrophilic dimple patterned superhydrophobic surfaces utilized in water droplet manipulations, and what specific manipulations are mentioned in the abstract?

Comments on the Quality of English Language

Language has to be corrected by native speaker

Reviewer 2 Report

Comments and Suggestions for Authors

I have read the article "Chemical Instability Induced Wettability Patterns on Superhydrophobic Surfaces" with great interest. In this work, the authors developed superhydrophobic-hydrophilic patterns using HCl solution exposure techniques. However, specific crucial aspects require additional attention. Hence, I recommend that the publication of this work be considered after the authors address these significant details.

1.      Hydrophilic-superhydrophobic patterns could be employed for rewritable patterns and water harvesting applications. Authors are encouraged to discuss these important points in the introduction section and cite relevant references, for example: a) Efficient Water Collection on Integrative Bioinspired Surfaces with Star-Shaped Wettability Patterns. Adv.Mater.2014, 26, 5025–5030. b) Synergistic chemical patterns on a hydrophilic slippery liquid infused porous surface (SLIPS) for water harvesting applications. Journal of Materials Chemistry A 8 (47), 25040-25046. c) Substrate-Independent and Re-Writable Surface Patterning by Combining Polydopamine Coatings, Silanization, and Thiol-Ene Reaction. Adv. Funct. Mater.2021, 31, 2107716. d) Robust and self-healable bulk-superhydrophobic polymeric coating. Chemistry of Materials 29 (20), 8720-8728.

2.      How does 1H,1H,2H,2H-Perfluorodecyltriethoxysilane bind with the Cu rods? Please provide the FTIR spectra.

3.      In Figures 5 and 6, during the transfer of the water droplet, a slight volume loss is observed at points B and C. It is recommended that the author quantifies this volume loss. How is this material useful for droplet manipulation applications considering the observed volume loss?

4.      What is the water contact angle (WCA) of the surface after HCl treatment?

5.      How small can this pattern be made possible? Authors are suggested to try with a volume of HCl droplet less than 1 µl.

6.      The novelty of this work is not very clear. Authors are suggested to create a comparative table illustrating various techniques with wettability patterns on superhydrophobic surfaces and position their work within that table.

8.      Is this material abrasion-resistant? Authors are suggested to conduct various physical abrasion tests.

9.      This methodology is surface-dependent. Can these superhydrophobic Cu rods be transferred to another substrate using scotch tape or PDMS?

10.  Is this material conductive? If so, it could be utilized for electrochemical reduction of carbon dioxide applications as well. Authors are suggested to mention this atleat in the conclusion part for the future perspective of this material.

Comments on the Quality of English Language

 Minor editing of English language required

Round 2

Reviewer 2 Report

Comments and Suggestions for Authors

All the comments have been adequately addressed. This paper is suitable for publication.